# Inhibition of Gastrin-Releasing Peptide Attenuates Phosphate-Induced Vascular Calcification

**DOI:** 10.3390/cells9030737

**Published:** 2020-03-17

**Authors:** Hyun-Joo Park, Yeon Kim, Mi-Kyoung Kim, Jae Joon Hwang, Hyung Joon Kim, Soo-Kyung Bae, Moon-Kyoung Bae

**Affiliations:** 1Department of Oral Physiology, BK21 PLUS Project, School of Dentistry, Pusan National University, Yangsan 50610, Korea; phj3421@hanmail.net (H.-J.P.); graceyeon88@gmail.com (Y.K.); eenga@naver.com (M.-K.K.);; 2Dental and Life Science Institute, School of Dentistry, Pusan National University, Yangsan 50610, Korea; skbae@pusan.ac.kr; 3Department of Oral and Maxillofacial Radiology, School of Dentistry, Pusan National University, Dental Research Institute, Yangsan 50610, Korea; softdent@pusan.ac.kr; 4Department of Dental Pharmacology, BK21 PLUS Project, School of Dentistry, Pusan National University, Yangsan 50610, Korea

**Keywords:** gastrin-releasing peptide, gastrin-releasing peptide receptor, vascular smooth muscle cells, vascular calcification

## Abstract

Vascular calcification is the pathological deposition of calcium/phosphate in the vascular system and is closely associated with cardiovascular morbidity and mortality. Here, we investigated the role of gastrin-releasing peptide (GRP) in phosphate-induced vascular calcification and its potential regulatory mechanism. We found that the silencing of GRP gene and treatment with the GRP receptor antagonist, RC-3095, attenuated the inorganic phosphate-induced calcification of vascular smooth muscle cells (VSMCs). This attenuation was caused by inhibiting phenotype change, apoptosis and matrix vesicle release in VSMCs. Moreover, the treatment with RC-3095 effectively ameliorated phosphate-induced calcium deposition in rat aortas ex vivo and aortas of chronic kidney disease in mice in vivo. Therefore, the regulation of the GRP-GRP receptor axis may be a potential strategy for treatment of diseases associated with excessive vascular calcification.

## 1. Introduction

Vascular calcification refers to the pathological deposition of calcium and phosphate minerals in the vascular system [1]. It is closely associated with aging, atherosclerosis and metabolic disorders, such as diabetes mellitus and chronic kidney disease [2,3]. Recent studies have shown that elevated phosphate levels are important for inducing vascular calcification, in both clinical trials and experimental models [4,5]. Potential mechanisms of phosphate-induced vascular calcification are apoptosis, matrix vesicle release and osteogenic/chondrogenic conversion of vascular smooth muscle cells (VSMCs) [6,7,8], making vascular calcification is a tightly regulated process, similar to bone formation. 

Gastrin-releasing peptide (GRP), a member of bombesin-like peptides, has been clearly implicated in several physiological and pathological processes, including exocrine and endocrine secretions, gastrointestinal tract motility, smooth muscle contraction and cell proliferation in normal and cancerous tissues [9,10]. The effects of GRP are mediated through the GRP receptor, one of the G protein-coupled receptors, that expressed in various cell types [11]. Increasing evidence has suggested that GRP and its receptor signaling promote endothelial dysfunction and migration and proliferation of VSMCs, which leads to the development of atherosclerosis [12,13]. In addition, GRP receptor signaling participates in the pathogenesis of inflammatory diseases such as sepsis and arthritis [14]. Inflammation is closely associated with vascular calcification, such that immune cells infiltrate plaque and release cytokines that regulate calcification [15]. However, the role of GRP/GRP receptor signaling in vascular calcification remains poorly understood.

Synthetic antagonists are designed to bind with high affinity to the receptors, thereby blocking signaling pathways. Recently, a nonpeptide antagonist for GRP receptor, RC-3095, was developed [16]. RC-3095 has been shown to possess anti-inflammatory properties in various models of inflammation in various experimental models of arthritis, gastritis, uveitis and sepsis [17,18,19] but its pharmacological action in vascular calcification has not been defined yet. 

In the present study, we investigated the effect of GRP inhibition on phosphate-induced calcification in VSMCs in vitro, an arterial ring ex vivo and the aorta of chronic kidney disease (CKD) mice in vivo and elucidated the underlying mechanism.

## 2. Materials and Methods

### 2.1. Reagents and Antibodies

RC-3095 and GRP were purchased from Sigma (St. Louis, MO, USA). The antibodies against GRP and GRP receptor were supplied by Abcam (Cambridge, UK) and Santa Cruz Biotechnology, (Santa Cruz, CA, USA), respectively. The antibodies against phospho-Smad1/5, Runx2, phospho- ERK, ERK, phospho-p38, p38, total/cleaved caspase-3, total/cleaved caspase-9, Bcl2 and Bad were obtained from Cell Signaling (Danvers, MA, USA). Calponin and β-actin antibodies were purchased from Abcam and Bioworld Technology (St. Louis Park, MN, USA), respectively. Smad1/5, horseradish peroxidase-conjugated goat anti-rabbit and anti-mouse IgG were obtained from Thermo Fisher Scientific (Waltham, MA, USA).

### 2.2. Cell Isolation and Culture 

To isolate VSMCs, male Sprague–Dawley rats (3 weeks old, 40–60g, Samtaco, Osan-si, Gyeonggi-do, Korea) were euthanized using intraperitoneal injection of sodium pentobarbital (60 mg/kg). The thoracic aorta was cut out and the surrounding fat and connective tissues were discarded. It was slit longitudinally, and its lumen surface was scraped with a razor blade to remove the intima, before cutting it into 3–5 mm long pieces. It was explanted lumen side down on collagen-coated culture dishes. After seven days, tissue fragments were discarded and sprouted VSMCs were collected (referred to as P0). A7r5 cells, purchased from the American Type Culture Collection (ATCC, CRL-1444^TM^) and primary VSMCs were grown in Dulbecco’s modified Eagle’s medium (DMEM, Thermo Fisher Scientific) with 10% fetal bovine serum (FBS, Thermo Fisher Scientific) and 1% antibiotics (Thermo Fisher Scientific), at 37 °C in 95% humidified air with 5% CO_2_. 

### 2.3. Calcification Induction and Quantification

A solution of inorganic phosphate (Pi) (Na_2_HPO_4_ and NaH_2_PO_4,_ pH 7.4) was added to serum supplemented-DMEM at concentrations of 1.4, 2.6 and 3.5 mM (calcification medium). After the indicated incubation period, the cellular calcium content and alkaline phosphatase (ALP) activity were determined using the calcium colorimetric assay kit (BioVision, Milpitas, CA, USA) and ALP assay kit (Takara, Los Angeles, CA, USA), respectively. For protein extraction, cells were solubilized in 0.1 M NaOH with 0.1% SDS and their protein content was measured by a Bio-Rad protein assay. 

### 2.4. Alizarin Red S Staining

To observe calcium deposition, cells grown on plastic supports were fixed with 4% paraformaldehyde and stained with 1 mg/mL alizarin red S solution, prepared by dissolution in deionized water adjusted to pH 4.1 to 4.3 with 10% NH_4_OH, for 30 min at 37 °C. Samples were rinsed and the stained calcium deposits were photographed. Once micrographs were captured, calcium deposits were destained and dissolved in 10% acetic acid. Absorbance at 420 nm was measured using a multi-detection microplate reader (Dynex, Lincoln, UK) to quantify calcification.

### 2.5. Von Kossa Staining

After dewaxing in xylene and rehydrating through a graded alcohol series, slides carrying tissue slices were soaked in distilled water, before incubating in 1% silver nitrate solutions under a UV lamp for 15 min, followed by soaking in 5% sodium thiosulfate for 5 min and cleaning with distilled water. The sections were then incubated with nuclear fast red for 5 min at room temperature and dehydrated with graded alcohol.

### 2.6. Analysis of Calcification

The photo micrographic images of stained calcium depositions were processed with a software developed in-house using MATLAB 2019 (Math Works), following a method described by Jonkman et al. [20]. In the first step, the images were differentiated into object and background, based on their intensity cut-off value interactively, followed by "flat-field correction" with interactive parameter adjustment. The correction uses Gaussian smoothing with a standard deviation of sigma (default 30) to approximate the shading. Finally, the objects were automatically segmented using the expectation maximization algorithm [21].

### 2.7. Enzyme-Linked Immunosorbent Assay (ELISA)

The amounts of secreted GRP protein in conditioned medium were determined by ELISA according to the manufacturer’s instructions (Cusabio, Houston, TX, USA). The absorbance of the samples at 450 nm was measured using an ELISA reader (Dynex) and the GRP levels were determined by interpolating the values on to a standard curve generated as per the manufacturer’s instructions.

### 2.8. Quantitative Real-Time RT-PCR

Real-time RT-PCR quantification was performed using an SYBR^Ⓡ^ Green method (Roche Applied Science, Penzberg Upper Bavaria, Germany). Cycling parameters included 1 cycle at 95 °C for 10 min, followed by amplification for 30 cycles at 95 °C for 10 s, 57 °C for 5 s and 72 °C for 7 s. The entire cycling process, including data analysis, took less than 60 min and was monitored using Light Cycler software (version 4.0). The sequences of oligonucleotide primers for real-time RT-PCR are listed in the Appendix A.

### 2.9. Western Blot Analysis

Harvested cells were lysed in RIPA buffer with a protease inhibitor cocktail (Sigma-Aldrich) and the protein concentration was measured by the BCA assay. Equal amounts of protein (30 μg/lane) were separated using SDS-PAGE and then transferred to a nitrocellulose membrane (GE Healthcare Life Sciences, Lafayette, CO, USA). The membrane was blocked with 5% skim milk in TBS containing 0.1% Tween 20, for 60 min and probed with appropriate antibodies. The signal was detected using enhanced chemiluminescence (ECL) reagent (GE Healthcare Life Sciences).

### 2.10. Gene Knockdown by Small Interfering RNA

The small interfering RNA (siRNA) duplexes for rat *GRP* and a negative control siRNA were obtained from GenePharma. The A7r5 cells were transfected using Amaxa nucleofector (Lonza, Basel, Switzerland) according to the manufacturer’s instructions.

### 2.11. Immunocytochemistry

Cells cultured on a coverslip were fixed in 4% paraformaldehyde for 10 min, blocked with 0.5% Triton X-100 in PBS for 5 min and then incubated first with appropriate primary antibodies followed by Alexa^®^ 488 and 594-conjugated secondary antibodies. Coverslips were mounted in Vectastain containing DAPI (Vector Laboratories, Burlingame, CA, USA). Results were analyzed using fluorescence microscopy (Nikon, Minato, Tokyo, Japan).

### 2.12. Flow Cytometry Analysis

VSMCs and A7r5 cells were seeded in 60 mm dishes and incubated overnight at 37 °C. They were incubated in calcification medium for 5 days, washed twice in 1× PBS and fixed in chilled 70% ethanol. The cells were stained with 5 μg/mL propidium iodide (PI, Sigma) at 23–25 °C for 10 min and analyzed by FACS Calibur (BD Bioscience, San Jose, CA, USA). The cell cycle profile was determined using the Modfit LT software.

### 2.13. TUNEL Assay

Apoptotic cells were detected using the Dead EndTM Fluorometric TUNEL System (Promega, Madison, WI, USA) in accordance with the manufacturer’s instructions. Cells were incubated in calcification medium for 5 days, fixed in 4% paraformaldehyde for 25 min at 4 °C and permeabilized with 0.2% Triton X-100 for 5 min at room temperature. Free 3′ ends of fragmented DNA were enzymatically labelled with the TdT-mediated dUTP nick end labelling (TUNEL) reaction mixture for 60 min at 37 °C in a humidified chamber. Labelled DNA fragments were visualized under a fluorescence microscope (Nikon, Minato, Tokyo, Japan).

### 2.14. Matrix Vesicle Isolation

Matrix vesicles were harvested using a modified matrix vesicle isolation protocol [22]. Confluent VSMCs were washed twice with PBS and transferred to control or calcification medium for 5 days, digested with collagenase and centrifuged at 10,000× *g* to remove cells and apoptotic bodies. Matrix vesicles were then harvested from the supernatant after centrifugation at 100,000× *g* for 30 min at 4 °C in an ultracentrifuge (Beckman, Brea, CA, USA). Th matrix vesicles were then resuspended with 1% Triton X-100 and protein and ALP activity were determined. 

### 2.15. Arterial Ring Calcification

Aortas (from the thoracic to the iliac arteries) were removed in a sterile manner from male Sprague–Dawley rats (6 weeks old, 150–200 g, Samtaco). After the adventitia and endothelium were carefully removed, the vessels were cut into 2–3mm rings and placed in either calcification medium or normal culture medium at 37 °C under 5% CO_2_ for 10 days, with medium changes once every 3 days.

### 2.16. Induction of Chronic Kidney Disease (CKD) in the Mice Model

Eight-week-old male C57BL/6 (20–25 g, Samtaco) mice were randomly assigned to the following experimental groups, with 5–10 animals in each group (Appendix A): the sham-normal phosphate (NP) group, the sham-high phosphate (HP) group, the CKD-NP group, the CKD-HP group and the CKD-RC-3095 treatment group. The animals from sham groups were fed a normal diet, while those from CKD groups were fed a diet supplemented with 0.2% (*w/w*) adenine. Mice in NP groups and HP groups were fed with pellet chow containing 0.5% phosphate and 1.8% phosphate (Jackson Laboratory), respectively, for 12 weeks [23]. For RC-3095 treatment, CKD mice were injected intraperitoneally with RC-3095 (1 mg/kg) or vehicle (0.9% NaCl), three times a week for 12 weeks. The mice were weighed once a week during this period. All animals were sacrificed with CO_2,_ on the last day of their feeding period. Blood and aortas were collected for further analysis. 

All animal studies (Section 2.2, Section 2.15 and Section 2.16) were conducted in accordance with the Guide for the Care and Use of Laboratory Animals (NIH publication No. 85–23 revised 1996) and were approved by the Institutional Animal Care and Use Committee at Pusan National University, Korea.

### 2.17. Statistical Analysis

Data shown are the mean ± standard deviation (S.D.), obtained from at least three independent experiments. Statistical comparisons between groups were made by one-way analysis of variance (ANOVA) followed by a Student’s t-test.

## 3. Results

### 3.1. Elevated Expression of GRP and GRP Receptor in Pi-Induced VSMC Calcification

Incubation of VSMCs in calcification medium with higher concentrations of Pi resulted in calcium deposition in a concentration-dependent manner within 8 days, whereas at a concentration equivalent to the human physiological serum phosphate level (1.4 mM) [24], Pi did not induce calcium deposition in the same time period (Figure 1a). Quantitative analysis indicated that as compared to normal conditions, calcium content of the VSMCs significantly increased under calcifying conditions (Figure 1b). ALP activity, a vital marker of calcification, was also markedly elevated in the presence of 2.6 mM and 3.5 mM Pi (Figure 1c). Next, to investigate the role of GRP in triggering VSMC calcification, we estimated the levels of GRP in calcified VSMCs using ELISA and found that the amount of secreted GRP proteins progressively increased with the increasing severity of VSMC calcification (Figure 1d). In addition, western blot analysis showed that GRP and GRP receptor expression levels significantly increased in the VSMCs under calcifying medium (Figure 1e). The levels of GRP and GRP receptor mRNA in Pi-induced VSMC calcification was further analyzed by real-time RT-PCR. Consistent with the western blot, the levels of GRP and GRP receptor mRNA were significantly elevated in the calcified VSMCs (Figure 1f). Next, we checked whether exogenous amidated GRP_1–27_ augments the calcification of VSMCs induced by Pi in VSMCs. As shown in Appendix A, calcification occurred upon Pi treatment at 2.6 and 3.5 mM as expected. However, cotreatment of GRP_1-27_ had little effect on Pi-induced VSMC calcification. 

### 3.2. Inhibition of GRP Attenuates Pi–Induced Osteogenic Differentiation of VSMCs

To evaluate whether inhibition of GRP attenuates Pi-induced VSMC calcification, we employed RC-3095, a GRP receptor antagonist and *GRP* gene knockdown by siRNA. The silencing effect of GRP siRNA transfection was confirmed by a decrease in the protein and mRNA expression level of GRP (Appendix A). As shown in Figure 2a,b and Appendix A, both GRP silencing and RC-3095 treatment attenuated calcium deposition and decreased calcium content of VSMCs, even under calcifying conditions. ALP activity also showed that both GRP silencing and RC-3095 treatment markedly inhibited calcification in VSMCs (Figure 2c). To further investigate the effect of GRP inhibition on the Pi-induced osteoblastic phenotype change of VSMCs, we examined the expression of runt-related transcription factor 2 (Runx2), a specific osteogenic marker and calponin, a contractile phenotype maker. Treatment of VSMCs with increasing concentrations of Pi (Appendix A) led to a dose-dependent increase in the expression of Runx2 protein, whereas GRP silencing (Figure 2d) and treatment with RC-3095 (Figure 2d,e and Appendix A) inhibited this increase. In contrast, the calponin protein level decreased significantly in VSMCs treated with high concentrations (2.6 mM and 3.5 mM) of Pi (Appendix A) and this was completely reversed in the presence of GRP siRNA and RC-3095 (Figure 2d and Appendix A). In addition, the immunofluorescence assay revealed a higher amount of calponin protein and a lower level of Runx2 protein in both GRP-silenced and RC-3095-treated groups (Figure 2e). Quantitative real-time RT-PCR analysis confirmed that both GRP-silencing and RC-3095 treatment upregulated the mRNA expression of calponin and downregulated that of Runx2 during Pi-induced osteogenic differentiation in VSMCs (Figure 2f). Phosphorylated Smad1/5 forms a complex with Smad4 and then moves to the cell nucleus where it recruits cofactors and Runx2 to regulate the expression of osteogenic genes like Runx2 and Osterix [25]. Both the ERK/MAPK pathway, as well as Smad signaling pathways, converge at transcription factors, for example, Runx2 to promote osteoblast differentiation [26]. We found that the treatment of Pi induced significant phosphorylations of Smad1/5, ERK1/2 or p38MAPK (Appendix A) and these increase were blocked by treatment with GRP siRNA or RC-3095 (Figure 2d and Appendix A).

### 3.3. Inhibition of GRP Ameliorates Pi–Induced Apoptosis and Matrix Vesicle Release of VSMCs

Since apoptosis of VSMCs is a major cause of vascular calcification [27], we assessed the extent of apoptosis using propidium iodide staining and TUNEL assay in calcified VSMCs. As expected, Sub-G1 peaks were enhanced in Pi-treated VSMCs, suggesting extensive apoptosis of these cells (Figure 3a). Furthermore, *GRP* knockdown or RC-3095 treatment substantially inhibited Pi-induced apoptosis in VSMCs, with apoptotic cells decreasing by 18.8% or 24.5%, respectively (Figure 3a). Likewise, we found that TUNEL-positive apoptotic cells markedly decreased when VSMCs were treated with either GRP siRNA or RC-3095 (Figure 3b). Next, we checked the expression levels of various pro-apoptotic and anti-apoptotic markers in calcified VSMCs induced by different concentration of Pi (Appendix A). The levels of cleaved caspase-3, cleaved caspase-9 and Bad increased by treatment with 2.6 mM Pi and decreased by *GRP* knockdown in VSMCs through siRNA or RC-3095 treatment (Figure 3c and Appendix A). We also found that these treatments strongly induced the expression of Bcl2 protein in calcified VSMCs (Figure 3c and Appendix A). Increasing evidences indicate that the release of matrix vesicle is the starting point of vascular calcification [28]. We confirmed that the extent of matrix vesicle release was directly dependent on the amount of Pi present in the medium (Appendix A). The release of matrix vesicles was stimulated by more than 8-fold in the presence of 2.6 mM Pi compared to controls (Appendix A) and treatment with GRP siRNA and RC-3095 significantly inhibited Pi-induced matrix vesicle release (Figure 3d). In addition, the matrix vesicles released from Pi-treated VSMCs showed increased ALP activity (Appendix A), which is a marker of matrix vesicle maturation [29], whereas *GRP* knockdown and RC-3095 treatment blocked this enzyme activity in the matrix vesicles (Figure 3e and Appendix A).

### 3.4. RC-3095 Suppresses Vascular Calcification in Ex Vivo Aortic Culture

To investigate whether Pi-stimulation induces vascular calcification ex vivo, pieces of thoracic rat aorta were cultured with different concentrations of Pi for 10 days [30]. Under normal phosphate conditions, no calcification was observed in organ cultured aortas; however, vascular smooth muscle layers of the aorta showed calcium deposition in the presence of 2.6 mM and 3.5 mM Pi (Figure 4a). The extent of the calcified area and expression of GRP and GRP receptor also increased in aortic explants upon treatment with these doses of Pi. (Figure 4a,b). Using real-time RT-PCR, we quantified GRP and GRP receptor mRNA and found that Pi enhanced the expression of both, in a dose-dependent manner (Figure 4c). Calcium deposition was strongly inhibited, when calcified aortas were treated with RC-3095 (Figure 4d). The extent of the calcified area also decreased in the presence of RC-3095. Western blot analysis revealed that RC-3095 treatment significantly reversed Pi-induced reduction of calponin and Bcl2, as well as enhancement of Runx2 and Bad (Figure 4e,f).

### 3.5. RC-3095 Alleviates Aortic Calcification In Vivo

The effect of RC-3095 on vascular calcification was investigated in vivo using a chronic kidney disease (CKD) mouse model fed a high phosphate (HP) diet [31]. Between-group comparisons in body weight and blood biochemical parameters are summarized in Appendix A. As shown in Figure 5a and Appendix A, considerable calcium deposition was seen only in CKD + HP mice, whereas all other groups (NP-, HP- sham mice or CKD + NP mice) showed no calcium deposition. The levels of GRP and GRP receptor mRNA were significantly elevated in the CKD + HP group (Figure 5b). In addition, we observed that GRP levels in plasma and the expressions of GRP and GRP receptor in aorta are higher in rat CKD models than sham rats (Appendix A). As shown in Figure 5c, the periodic injection of RC-3095 decreased the formation of arterial medial calcification in CKD+HP mice. The extent of the calcified area also decreased significantly when CKD + HP mice were treated with RC-3095 (Appendix A). Moreover, the expression of GRP and GRP receptor mRNA increased in the thoracic aorta of the CKD + HP mice and it was restored to normalcy by RC-3095 treatment (Figure 5d). Quantitative real-time RT-PCR demonstrated that RC-3095 treatment blocked the osteogenic conversion and apoptosis in the aorta of CKD+HP mice (Figure 5e).

## 4. Discussion

Endothelial cells and VSMCs are associated with the pathological processes involved in atherosclerosis, including endothelial dysfunction, vascular inflammation, VSMC proliferation and migration and vascular calcification [32,33]. At the early stage of atherosclerosis, cardiovascular risk factors activate the vascular endothelium and VSMCs, leading to the expression of chemokines, cytokines and adhesion molecules that interact with leukocytes [34,35]. At later stages of atherosclerotic plaque development, VSMCs proliferate and migrate from the vessel wall into the lesion, where they secrete an extracellular matrix, resulting in the formation of atherosclerotic plaques with a fibrous cap [32]. In the advanced stages, calcification and neovascularization contribute to the progression and rupturing of the atherosclerotic plaques, leading to the instability of atherosclerotic lesions [36,37]. Interestingly, our previous findings proved that GRP directly activates vascular endothelial cells and subsequently increases monocyte-endothelial adhesion through the GRP receptor [12]. We also demonstrated that GRP significantly enhances the proliferation and migration of VSMCs by increasing the expression of matrix metalloproteinase-2 and -9 [13]. In addition, it has been reported that GRP acts as a proangiogenic factor by binding to the GRP receptor, present in endothelial cells [38]. In the present study, we demonstrated that inhibition of GRP attenuates calcium deposition, phenotypic transition and apoptosis in calcified VSMCs. Therefore, these findings, together with previous observations, suggest that the GRP-GRP receptor axis may be involved in the progression of atherosclerosis by endothelial dysfunction, neovascularization, proliferation and migration of VSMCs and vascular calcification.

A link has been found between the apoptosis of VSMCs and vascular calcification, suggesting that apoptosis is one of critical events for vascular calcification [39]. Recent evidences have shown that apoptosis promotes matrix calcification, primarily through the release of calcifying, membrane-bound matrix vesicles, such as apoptotic bodies, which act as nucleation sites for calcification in blood vessels [40]. In fact, apoptosis occurs in VSMCs before the onset of calcification and VSMC “blebs” or apoptotic bodies probably concentrate calcium in a crystallized form [41]. Increasing evidence also show that VSMCs undergoing osteogenic transformation promote calcification by releasing matrix vesicles capable of nucleating hydroxyapatite [42]. Zhang et al. [43] demonstrated that *Sp1* gene silencing delayed vascular calcification by inhibiting the processes of phenotype switch, apoptosis and matrix vesicle release in VMSCs. Therefore, it is possible that blocking apoptosis and matrix vesicle release in calcifying VSMCs ameliorates vascular calcification. Here, we found that GRP silencing as well as RC-3095 treatment strongly attenuated apoptosis and matrix vesicle release, which alleviated the calcification of VSMCs and rat arterial rings. Meanwhile, RC-3095 treatment was effective in reducing vascular apoptosis in vivo as well. Thus, the inhibitory effects of GRP silencing and RC-3095 on the initiation and progression of vascular calcification may lie in their ability to prevent the apoptosis of VSMCs.

Vascular calcification is a typical pathological feature of chronic kidney disease (CKD) and it contributes to cardiovascular morbidity and consequent mortality in patients with CKD [44]. Although high phosphate has been identified as a key risk factor for vascular calcification in the CKD population, the mechanisms of vascular calcification are not completely understood and the current therapies for CKD have limited efficacy [45]. Thus, there is a need to identify new agents that target key molecular pathways involved in the pathogenesis of vascular calcification. In the present study, we used an adenine-based mouse model of CKD with a range of severity of the vascular calcification, with levels of blood urea nitrogen, creatinine, phosphate and calcium comparable to those reported previously [23,46]. Here, we found that the increased expression of GRP and GRP receptor was involved in HP-induced vascular calcification in CKD mice and RC-3095 treatment significantly attenuated the arterial medial calcification in them, making GRP a potential therapeutic target for vascular calcification in CKD. Although the adenine-based mouse model of CKD is commonly used for the study of vascular calcification, an adenine-rich diet causes weight loss (Appendix A) and faster development of vascular calcification in mice than in humans [47]. Therefore, the pathogenic role of GRP, if any, in vascular calcification in patients with CKD and the possible strain differences found in various mouse models [48] need to be carefully considered before targeting GRP in the treatment of human CKD. 

The process of vascular calcification is actively regulated by various inducers and inhibitors, which are possibly associated with and driven by developmental, inflammatory or metabolic factors [49]. TNF-alpha induces vascular calcification by promoting ALP and the Wnt-β-catenin signaling pathway like hyperphosphatemia [50,51]. Disturbance or downregulation of negative regulators like Fetuin-A lead to the phenotypic transformation of VSMCs, which leads to VSMC mineralization [52]. Also, oxidative stress and oxidized lipids from hyperlipidemia or an abnormal dose of vitamin D promotes VSMC calcification by upregulating Runx2 [53,54,55]. Further investigations will be necessary to determine the crucial role of GRP inhibition in vascular calcification driven by the perturbation of diverse modulators promoting or inhibiting calcification.

RC-3095 has been developed as anticancer candidate compounds, exhibiting impressive anti-cancer activity both in vitro and in vivo in various murine and human tumors [56]. Also, RC-3095 has been shown to have anti-inflammatory properties in murine models of arthritis, gastritis, uveitis and sepsis [17,18,19]. RC-3095 has been reported to exert anti-inflammatory and immunomodulatory effects in different rat and mouse models of arthritis [19]. We demonstrated here the inhibitory effect of RC-3095 on vascular calcification in rat VSMCs in vitro, in rat aortic rings ex vivo and in aortas from the mouse CKD model in vivo. Additionally, we confirmed that the plasma GRP levels and expression levels of GRP and GRP receptor in aorta are upregulated in rat CKD models. However, it is necessary to demonstrate the effect of RC-3095 on aortic calcification in a rat CKD models.

In conclusion, our results provide the first evidence of the role of GRP inhibition on the amelioration of phosphate-induced vascular calcification by inhibiting matrix vesicle release, apoptosis and osteogenic differentiation of VSMCs in vitro and vascular calcification of the arterial ring ex vivo and that of the aortas of CKD mice in vivo. These findings provide important clues regarding the mechanisms of vascular calcification and suggest that targeting the GRP-GRP receptor axis may be an attractive strategy for treatment of diseases associated with excessive vascular calcification. 

## Figures and Tables

**Figure 1 cells-09-00737-f001:**
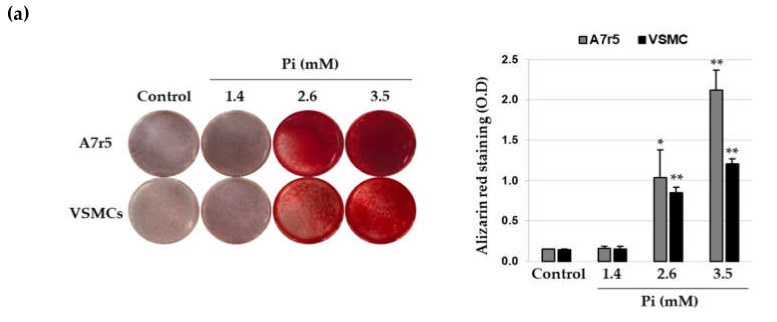
Expression of gastrin-releasing peptide (GRP) and GRP receptor during Pi-induced calcification in vascular smooth muscle cells (VSMCs). A7r5 and primary VSMCs were cultured in calcification medium (1.4, 2.6 and 3.5 mM Pi) for 7 days. (**a**) Calcium deposition was analyzed by staining with ARS (left), followed by measurement of absorbance to evaluate the degree of mineralization (right). * *p* < 0.05; ** *p* < 0.01 vs. control. (**b**) Calcium content was measured by colorimetric calcium assay. * *p* < 0.05; ** *p* < 0.01 vs. control. (**c**) ALP activity was measured and normalized to protein content, for quantitative analysis. * *p* < 0.01 vs. control. (**d**) Secreted GRP content in the conditioned cell culture medium was measured using ELISA. * *p* < 0.01 vs. control. (**e**) GRP and GRP receptor (GRP-R) protein levels were examined by western blotting using specific antibodies. β-actin served as the loading control. (**f**) Total RNA was isolated and analyzed by real-time RT-PCR using the specific primers for rat *GRP* and *GRP-R* genes. The expression level of the control (untreated) was set to 1 and the values were normalized to the β-actin mRNA levels. * *p* < 0.01 vs. control. Data shown are the mean ± SD, obtained from at least three independent experiments.

**Figure 2 cells-09-00737-f002:**
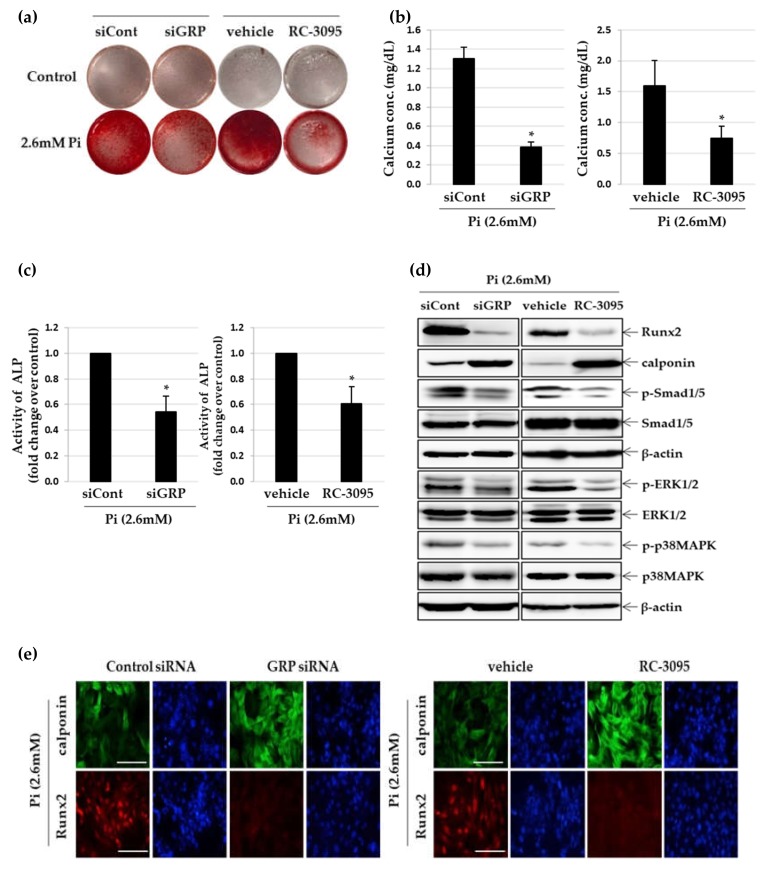
Effect of GRP silencing and RC-3095 treatment on Pi-induced osteogenic differentiation in VSMCs. A7r5 cells were transfected with GRP siRNA or negative control siRNA for 24 h and cultured in calcification medium (2.6 mM). Primary VSMCs were cultured in calcification medium with or without RC-3095 (1 μM). After 5 days in culture, VSMC calcification was determined by ARS staining (**a**), calcium content assay (**b**) and ALP activity assay (**c**). * *p* < 0.01 vs. control siRNA or vehicle. (**d**) Western blots were individually probed with antibodies against Runx2, calponin, *p*-Smad1/5, Smad1/5, *p*-ERK1/2, ERK1/2, *p*-*p*38MARK, *p*38MAPK or β-actin. (**e**) Expression of Runx2 (red) and calponin (green) was examined by fluorescence immunocytochemistry using specific antibodies. Nuclei were stained with DAPI (blue) (original magnification, **×**400). Scale bar: 50 μm. (**f**) Using real time RT-PCR, the expression levels of Runx2 and calponin mRNA were quantified. * *p* < 0.01vs. control siRNA or vehicle. Data shown are the mean ± SD, obtained from at least three independent experiments.

**Figure 3 cells-09-00737-f003:**
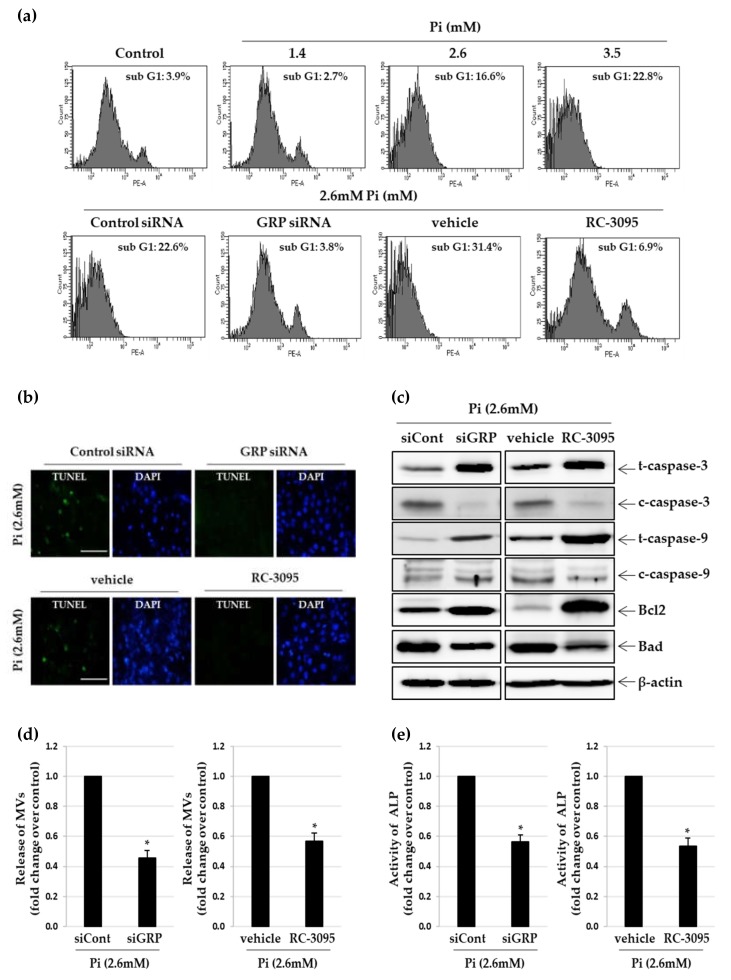
Effect of *GRP* knockdown and RC-3095 treatment on Pi-induced apoptosis and matrix vesicle release in VSMCs. A7r5 cells transfected with GRP siRNA or negative control siRNA for 24 h and cultured in calcification medium (2.6 mM) for 5 days. Primary VSMCs were cultured in calcification medium with or without RC-3095 (1 μM) for 5 days. (**a**) Induction of apoptosis was detected by flow cytometry with PI staining. (**b**) Apoptosis in A7r5 cells (green) was determined by TUNEL staining. Nuclei were stained with DAPI (blue) (original magnification, **×**400). Scale bar: 50 μm. (**c**) Western blots were individually probed with antibodies against total/cleaved-caspase-3, total/cleaved-caspase-9, Bcl2, Bad and β-actin. (**d** and **e**) Matrix vesicles were isolated as described in the Section 2. ALP activity was measured and normalized to the total matrix vesicle protein content. **p* < 0.01 vs. control siRNA or vehicle. Data shown are the mean ± SD, obtained from at least three independent experiments.

**Figure 4 cells-09-00737-f004:**
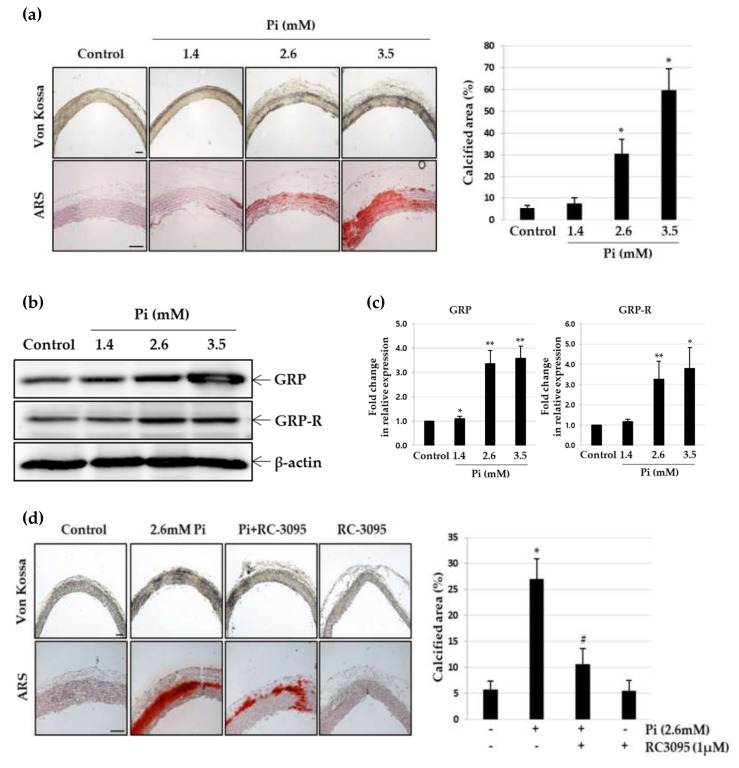
Effect of RC-3095 on Pi-induced vascular calcification in cultured explants of aorta. Pieces of rat aorta were cultured in calcification medium (2.6 mM) for 10 days. (**a**) The calcified lesions were examined by von Kossa and ARS staining (left). Scale bar: 50μm. The percent calcified area was calculated using Calcification Analyzer Ver2 (right). * *p* < 0.01 vs. control. (**b** and **c**) Levels of GRP or GRP-R proteins and their mRNA were estimated by western blot analysis and real-time RT-PCR, respectively. * *p* < 0.05; ** *p* < 0.01 vs. control. Pieces of rat aorta were incubated in calcification medium (2.6 mM) in the presence or absence of RC-3095 (1 μM) for 10 days. (**d**) The calcified lesions were examined by von Kossa staining (left). Scale bar: 50μm.The percent calcified area was calculated using Calcification Analyzer Ver2 (right). * *p* < 0.01 vs. control, ^#^
*p* < 0.01 vs. 2.6 mM Pi, Data shown are the mean ± SD, obtained from at least three independent experiments. (**e**,**f**) Western blots were individually probed with antibodies against Runx2, calponin, Bcl2, Bad or β-actin.

**Figure 5 cells-09-00737-f005:**
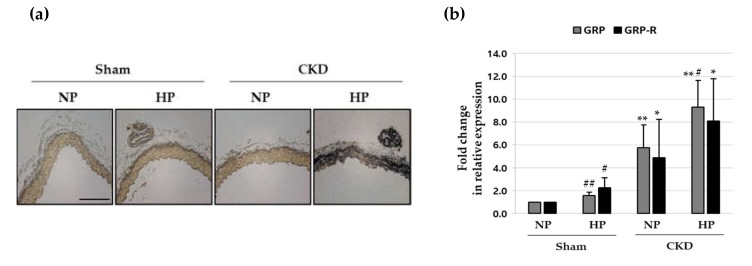
Effect of RC-3095 on vascular calcification in HP-fed CKD mouse. CKD mice were fed with a normal phosphate (0.5%, NP) or high phosphate (1.8%, HP) diet. For assessing the effect of RC-3095 treatment, CKD mice were intraperitoneally injected with RC-3095 (1 mg/kg) or vehicle (0.9% saline) 3 times a week for a total 12 weeks. (**a** and **c**) Area of aortic calcification in CKD and sham mice was determined by von-Kossa staining. Scale bar: 50 μm. (**b**) Total RNA was isolated and analyzed by real-time RT-PCR using the specific primers for mouse *GRP* and *GRP-R*. The expression level of these genes in the control (untreated) was set to 1 and the values were normalized to the β-actin mRNA levels. * *p* < 0.05; ** *p* < 0.01 vs. sham, ^#^
*p* < 0.05; ^##^
*p* < 0.01vs. NP. (**d**,**e**) GRP, GRP-R, Runx2, calponin, Bcl2 and Bad mRNA levels were also examined by real-time RT-PCR. * *p* <0.05; ** *p* < 0.01 vs. NP, ^#^
*p* < 0.05 vs. vehicle. Data shown are the mean ± SD, obtained from at least three independent experiments.

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
