# Peer review of "Inhibition of Gastrin-Releasing Peptide Attenuates Phosphate-Induced Vascular Calcification"

_cells, 2020, doi:10.3390/cells9030737_

Round 1

Reviewer 1 Report

Comments to Author:

In this manuscript authors describe the role of gastrin-releasing peptide (GRP) in phosphate-induced vascular calcification and indicate the regulatory pathway. Vascular calcification is common in chronic kidney disease (CKD) and the mechanism is multifactorial. The ever-growing evidence now linking to many other clinical conditions such as end stage renal disease. Moreover, multiple risk factors in patients with CKD can be predisposed for vascular calcification because of that induce vascular smooth muscle cells to change into a chondrocyte or osteoblast-like cell. In addition to dietary phosphate, high total body burden of calcium and phosphorus due to abnormal bone metabolism and the reduced levels of circulating and locally produced inhibitors are the factors that can increase risk and complicate the management of vascular calcification. Therefore, extensive biological study in animal model needs to be done before proceeding to use pharmacological agent, so it is not clear the overall contribution of Gastrin-releasing peptide (GRP) mediated pathway in vascular calcification and what extent such process can be reversed by GRP receptor antagonist RC-3095. Overall, the manuscript well planned but the rationale and conclusions need to be revised since they did not address the main aim and objective of the study and there is lack of clarity of what can be reversed.

Some more points to consider:

  1. Considering the in vivo condition, there may be off-target effects since they only show the effect of GRP receptor antagonist RC-3095, not any other similar (control compound).

  1. Moreover, the animal model described here as CKD and non-CKD not defined well, so it is confusing to understand phosphate-induced/non-induced vascular calcification.

  1. This study implements both cell line (A7r5 cells) and primary cell (VSMC from male Sprague-Dawley rats), which is commendable, however it would be nice to know about a non-phosphate-induced calcification.

  1. Moreover, using both pharmacological [antagonist (RC-3095) for the GRP-R] and genetic [GRP siRNA] for the cell line, are good, however, in siRNA work is that it was done only using 2.6 mM of Pi.

  1. How they know CKD was induced after feeding with adenine 0.2%. Is any supplemental material which has metabolic status and weight conditions can suggest such claim?

  1. Expression of some calcification and inflammatory genes may be helpful.

  1. CKD model used, however no kidney/urine status mentioned.

  1. pH of AR S stain and the buffer condition need to be stated.

Reviewer 2 Report

The authors examined whether the inhibition of gastrin-releasing peptides (GRPs) could retard vascular calcification (VC) using in vivo and in vitro studies. Overall, the idea is not without interest. However, I have several major suggestions for their work.

  1. As the authors stated, GRPs have been shown to have multiple physiological functions and play a role in different diseases including cancer and central nervous system ones. According to others, GRPs exhibit their actions through regulating inflammatory responses and oxidative stress in several cells. However, from the authors’ current manuscript, I cannot find any mechanistic investigation within; the phenotypic changes resulting from GRP receptor antagonists are well shown, but what causes these beneficial effects? RUNX2 and calponin are later osteoblastic markers, while apoptotic proteins are one of the late phenomenon of calcification, and they may not be specific mediators of GRP/GRP receptors. The authors should conduct more mechanistic study by showing the direct downstream effector(s) of GRP/GRP receptors, specifically in vascular smooth muscle cells.
  2. A major concern of this study is the inconsistent model application. The authors initially used rat vascular smooth muscle cells and rat arterial rings for experiments, but in vivo, they used mice model, which seems intrusive. This casts doubt on the coherence of their findings. I think they should use rat models of vascular calcifications to validate their in vitro findings; without this, their findings may be jeopardized.
  3. The distribution of GRP and GRP receptors in this study is another unanswered question. Apart from central nervous systems, GRP has been shown to be distributed mainly in gastrointestinal/pancreatic/splenic tissues, ovary/testis, and several endocrine organs, possibly reflect its endocrine/exocrine regulatory nature. However, it is unclear whether this duo is expressed in vascular walls, at least within a reasonable level. The authors need to first demonstrate their expression levels in vascular media in order to study its physiologic importance.
  4. Another concern is the relative contribution of GRP and GRP receptor during the pathologic changes of VC. The authors simultaneously examined intracellular GRP and GRP receptor expressions in vascular smooth muscle cells, and also the secreted GRP in culture media. However, it appears confusing that how this duo takes action within the vascular cells; is this an autocrine loop that propagates by itself during VC? In this case, the authors must also use GRP-binding antibody in all relevant experiments to demonstrate the efficacy of blocking GRP per se, as they only used GRP receptor antagonist at this moment.

Round 2

Reviewer 2 Report

The authors have partially responded to my comments previously. However, there seem to be more concerns at this time. Please see below.

  1. For additional mechanistic studies, the authors focus on the activation of Smad 1/5 and ERK/MAPK pathways, which are integral and should be applauded. However, they did not put these data into the manuscript and said these data were for reviewer response only. I sincerely think that these data should all be incorporated into the main text instead of being withhold; otherwise the manuscript content becomes rather meager. Please put the “figures for reviewer purposes” all into the revised manuscript.
  2. For in vivo rat model experiments, the authors provide some pilot data about uremic rat model and the serum levels of GRP and aortic levels of GRP/GRP receptor in the figure 2 for reviewer purpose. Please put these data in the main text as well. In addition, I recommend that the authors repeat the experiments of RC-3095 (as in the original Figure 5) in the rat model they establish here and present these data in the main text, instead of using the mice model data as it is currently. This will increase result coherence and consistency. Using explanations in the discussion is not adequate and unconvincing. The authors can request longer time for preparing and conducting these experiments.
  3. How did the authors explain that co-treatment of GRP did not influence/increase calcification extent (supplementary Figure S1b) but GRP knockdown in turn suppressed calcification severity (Figure 2)? Furthermore, in supplementary Figures 2d~2e, there seems to be no difference between control siRNA and GPR siRNA group regarding calcium content and multiple calcification markers, especially in light of such wide confidence interval of the high phosphate groups; but the authors wrote in the text that there was a difference when using GRP knockdown. These contradictory descriptions between text and figures arouses concern for data consistency. Please explain.
